# No Association between Gastrointestinal Rebleeding and DOAC Therapy Resumption: A Systematic Review and Meta-Analysis

**DOI:** 10.3390/biomedicines11020554

**Published:** 2023-02-14

**Authors:** Dániel Pálinkás, Brigitta Teutsch, Endre Botond Gagyi, Marie Anne Engh, Patrícia Kalló, Dániel S. Veres, László Földvári-Nagy, Nóra Hosszúfalusi, Péter Hegyi, Bálint Erőss

**Affiliations:** 1Centre for Translational Medicine, Semmelweis University, H-1085 Budapest, Hungary; 2Department of Gastroenterology, Military Hospital—State Health Centre, H-1134 Budapest, Hungary; 3Institute for Translational Medicine, Medical School, University of Pécs, H-7622 Pécs, Hungary; 4Selye János Doctoral College for Advanced Studies, Semmelweis University, H-1085 Budapest, Hungary; 5Department of Biophysics and Radiation Biology, Semmelweis University, H-1094 Budapest, Hungary; 6Department of Morphology and Physiology, Faculty of Health Science, Semmelweis University, H-1088 Budapest, Hungary; 7Department of Internal Medicine and Haematology, Semmelweis University, H-1088 Budapest, Hungary; 8Institute of Pancreatic Diseases, Semmelweis University, H-1085 Budapest, Hungary

**Keywords:** non-vitamin K antagonist oral anticoagulant (NOAC), gastrointestinal hemorrhage, therapy resumption, safety, efficacy

## Abstract

Background: There are recommendations for anticoagulation resumption after gastrointestinal bleeding (GIB), although data addressing this topic by direct oral anticoagulants (DOACs)-treated patients is lacking. We aim to determine the safety and efficacy of restarting DOACs after GIB. Methods: Studies that reported rebleeding, thromboembolic events, and mortality after restarting or withholding DOACs were selected. The systematic research was conducted in five databases (MEDLINE, EMBASE, CENTRAL, Web of Science, and Scopus). The random effect model was implemented to calculate the pooled odds ratio (OR). The ROBINS-I tool was used for risk of bias assessment, and the certainty of the evidence was evaluated with the GRADE approach. Results: Four retrospective cohort studies (1722 patients) were included in the meta-analysis. We did not find a significant increase in the risk of rebleeding in patients restarting DOACs after index GIB (OR = 1.12; 95% CI: 0.74–1.68). The outcomes of thromboembolic events and mortality data were not suitable for meta-analytic calculations. Single studies did not show statistically significant differences. Data quality assessment showed a serious overall risk of bias and very low quality of evidence (GRADE D). Conclusion: DOAC resumption after a GIB episode may not elevate the risk of rebleeding. However, the need for high-quality randomized clinical trials is crucial.

## 1. Introduction

The use of direct oral anticoagulants (DOACs) has rapidly increased in the last decade [1,2]. The prescription of DOACs for patients with newly diagnosed atrial fibrillation has already exceeded the traditional vitamin K antagonist (VKA) in developed countries [3]. In addition to stroke prevention in patients with non-valvular atrial fibrillation (NVAF), DOACs are also prescribed to prevent and treat venous thromboembolism, namely, deep vein thrombosis and pulmonary embolism. In contrast with VKAs, DOACs do not need regular laboratory monitoring, have fixed dosing, and have fewer drug–drug and drug–food interactions that facilitate their application and adherence.

DOACs are reported to have at least non-inferiority in efficacy and a better safety profile than VKAs [4,5,6,7]. The only exception is gastrointestinal bleeding (GIB), where evidence is conflicting. GIB occurs in approximately 1.38–3.54 events per 100 patients per year but shows a notable variation across the types and doses of DOACs [8,9,10].

Whether and when to restart anticoagulant therapy after a GIB event remains debated. Data from VKA studies show an elevated rebleeding risk but decreased thromboembolic risk after resuming anticoagulation therapy [11,12]. Clinical guidelines suggest anticoagulant therapy resumption for patients with high thromboembolic risk once hemostasis is achieved [13,14,15,16]. However, these recommendations are based on low-quality evidence from studies on VKAs, and their conclusions should not be extrapolated to DOACs.

Due to the uncertainty and possibly unjustified fear of recurrent bleeding, anticoagulant therapy is permanently interrupted in many patients after gastrointestinal hemorrhage [17], which results in a three-fold increase in thromboembolic risk [11].

We aim to determine the safety and efficacy of DOAC therapy resumption after a GIB event and systematically appraise the available evidence.

## 2. Materials and Methods

We report our meta-analysis following the updated Preferred Reporting Items for Systematic Reviews and Meta-Analysis (PRISMA) 2020 statement [18] (see Appendix A). The study was performed following the Cochrane Handbook’s recommendations for Systematic Reviews of Interventions, Version 6.1.0.18. [19,20]). The protocol was registered in advance on PROSPERO (registration number CRD42021284314) and implemented without deviation.

### 2.1. Search Strategy

We conducted a systemic search of the following databases: MEDLINE (via PubMed), EMBASE, Cochrane Central Register of Controlled Trials (CENTRAL), Web of Science, and Scopus, on 20 October 2021, and repeated it on 7 November 2022. The search key can be found in the Appendix A. There were no restrictions imposed. We did not exclude conference abstracts to identify any studies that had not been published before. We performed an additional manual screening on the reference list of the included articles and relevant reviews and completed a forward and backward citation search.

### 2.2. Study Selection and Data Collection

Two independent authors (DP, EG) performed study selection with the EndNote X9 (Clarivate Analytics, Philadelphia, PA, USA) reference management program. Disagreements were solved by an independent author (BT). A PICO (patient—intervention—control—outcome) framework was used to identify eligible studies. We planned to include randomized-controlled trials and cohort studies regarding patients hospitalized with gastrointestinal hemorrhage while taking DOACs (P). Articles were only found to be eligible if they reported restarting (I) or withholding (C) DOAC therapy after admission and assessed the outcome of mortality, rebleeding, and/or thromboembolic events (O).

After automatic and manual duplicate removal, study selection was carried out based on the title and abstract, then by full-text content adhering to the predefined eligibility criteria. For inter-reviewer reliability measurement, Cohen’s kappa coefficient (κ) was calculated after each step [21]. We contacted the authors via email to request additional information where relevant data were not reported.

Two independent reviewers (DP, EG) extracted the data using a standardized data collection form. Disagreements were resolved by arbitration by a third independent researcher (BT). The following data were collected from each included study: title, author, year of publication, study design, population characteristics (number of patients, age, gender, comorbidity, indication for DOAC therapy, type of DOACs and additional antithrombotic therapy), location of the bleeding source, in-patient-management of the index bleeding (endoscopic intervention, need for admission to intensive care unit, need for transfusion), length of follow-up, and time to DOAC resumption and outcomes (gastrointestinal rebleeding, thromboembolic event and all-cause mortality).

### 2.3. Risk of Bias Assessment

Two independent investigators (DP, EG) assessed the risk of bias using the ROBINS-I tool of Cochrane collaboration for non-randomized studies of intervention [20]. The following seven domains were evaluated: confounding, selection of participants in the study, classification of interventions, deviations from intended interventions, missing data, measurement of outcomes, and selection of the reported result. Any disagreement was resolved by a third author (BT). The risk-of-bias VISualization (robvis) web-based tool was used for drafting the figures [22].

### 2.4. Data Synthesis and Statistical Analysis

An odds ratio (OR) with a 95% confidence interval (CI) was used for the effect size measure. We anticipated considerable between-study heterogeneity, so a random-effects model was used to pool effect sizes. Pooled OR was calculated by the Mantel–Haenszel method [23,24]. To estimate the heterogeneity variance measure, the Paule–Mandel method was applied [24,25]. Additionally, between-study heterogeneity was described by Higgins and Thompson’s I2 statistics [26]. A forest plot was used to summarize the results graphically. The mentioned analyses and the forest plot were carried out with R software (R Core Team 2021, v4.1.1) [27] using the meta-package [28]. If data were available from less than three studies, meta-analytic calculations were not performed, and we reported the original data in the systematic review part.

Publication bias was not assessed due to the small study number.

### 2.5. Quality of Evidence

Certainty of evidence was assessed following the recommendation of the “Grades of Recommendation, Assessment, Development, and Evaluation (GRADE)” workgroup [29]. The summary of findings table was prepared with the GRADE profiler (GRADEpro) tool (GRADEpro Guideline Development Tool (Software)), McMaster University, 2020 (developed by Evidence Prime, Inc., Hamilton, ON, Canada) [30].

## 3. Results

### 3.1. Search and Selection

The systematic database search found 9188 articles. After duplicate removal, 6375 studies were screened, 9 of which fulfilled eligibility criteria, although 5 were excluded due to reporting on overlapping populations [31,32,33,34,35]. One article reported their results in a way that we could not include in the meta-analytic calculation [36]. Beyond the three remaining articles [37,38,39], one additional study was found through reference screening [40] (Figure 1).

### 3.2. Basic Characteristics of Included Studies

The baseline characteristics of the enrolled studies are detailed in Table 1. All the articles are observational retrospective cohort studies, three performed in the United States of America and one in Japan. In two studies [37,38], dabigatran, rivaroxaban, and apixaban were utilized; in one study [39], all DOAC types were the investigated anticoagulant treatments, while in the study by Hernandez et al. [40], besides dabigatran, warfarin-treated patients were also involved. From this article, only the data from DOAC-related GIB patients were extracted. Overall, 1722 patients were included in the meta-analysis. The average patient age was above 68 years. Indication for anticoagulation was atrial fibrillation in three articles [37,39,40] and multiple (atrial fibrillation, venous thromboembolism, and pulmonary embolism) in one study [38]. The source of index bleeding was detailed in three articles [38,39,40] as upper gastrointestinal (GI), lower GI, and unknown. The severity of index bleeding was described in three articles. In two studies [38,40], the International Society on Thrombosis and Haemostasis (ISTH) definition [41,42], and in one study [39], the Bleeding Academic Research (BARC) criteria was used [43]. Major bleeding occurred between 46 to 100% of cases. The definition of the intervention varied between therapy resumption in 7 days [38], 90 days [40], and during the follow-up time. It was a median of 40 days by Sengupta et al. [37], while Yanagisawa et al. [39] reported that 90% of patients restarted therapy in the first 14 days. The typical outcomes observed in all three articles were hospital readmission with GI bleeding or recurrent bleeding, while all-cause mortality [38], thromboembolic complications [37], and major adverse cardiac and cerebrovascular events (MACCE) [39] were investigated in one article. The follow-up period varied between an average of 6 to 28 months.

In addition to the article of Yanagisawa et al., no additional data were provided by the requested authors [39].

### 3.3. Gastrointestinal Rebleeding

Data from four studies [37,38,39,40], including 1,722 patients, were pooled in the meta-analytical calculation, and 53 out of 741 (7.15%) DOAC resumed, and 67 out of 981 (6.82%) DOAC-withheld patients suffered from GI rebleeding. The odds for recurrent GIB did not differ significantly in the case of DOAC therapy resumption compared with therapy withdrawal (OR = 1.12; CI: 0.74–1.68; I2 = 0%) (Figure 2).

### 3.4. Thromboembolic Event

The study by Sengupta et al. [37] reported thromboembolic complications, including 1338 individuals. The definition was 90-day hospital readmission with thromboembolism (TE), which contained venous and arterial TE, ischemic stroke, and transient ischemic attack. TE occurred in 16 out of 586 (2.73%) patients who restarted DOACs and 17 out of 752 (2.26%) patients who permanently interrupted the therapy. No significant difference was observed in the risk of thromboembolic complications between the two groups (HR = 0.98, CI: 0.37–2.21, *p* = 0.96).

Yanagisawa et al. [39] used major adverse cardiac and cerebrovascular events (MACCE) as a composite definition for thrombotic adverse events, including systemic thrombosis, myocardial infarction, and cardiac death. In this small study, 1 out of 45 patients who restarted DOAC therapy and three out of eight patients who did not resume anticoagulation suffered MACCE. The odds for MACCE were significantly lower in patients who restarted DOAC therapy (OR = 0.037, CI: 0.003–0.43).

### 3.5. All-Cause Mortality

In the article by Valanejad et al. [38], 4 out of 18 (22%) patients who restarted and 4 out of 37 (10.8%) patients who held DOAC therapy died during the follow-up period. All-cause mortality was not significantly higher in the DOAC resumption group (OR = 2.36, CI: 0.52–10.78).

### 3.6. Risk of Bias Assessment

Results are presented in Figure 3 and in the Appendix A). The quality assessment revealed a severe overall risk of bias in all outcomes and studies. Bias due to confounding was the main source of potential bias. DOAC therapy was more likely to be interrupted permanently by older patients with severe index bleeding, who needed more supportive treatment.

### 3.7. Quality of Evidence

Certainty of evidence proved to be very low in all three outcomes observed in this study; details are shown in the Summary of Findings table (Table 2).

## 4. Discussion

The decision to proceed with anticoagulant therapy resumption after GIB must be preceded by a risk stratification of thromboembolism and rebleeding. The impact of DOAC therapy restart on this complex condition is still unclear, as no previous analysis has summarized the data.

In our systematic review and meta-analysis, we evaluated the safety and efficacy of DOAC therapy resumption after an episode of gastrointestinal hemorrhage. Our results showed no association between restarting DOAC therapy and rebleeding. Data on mortality and thromboembolic complications were not sufficient for meta-analytic calculations. Single studies showed no association between these outcomes and therapy restart.

Previous meta-analyses addressing the topic of anticoagulant therapy resumption [11,12] included studies mainly with VKA-treated patients. Their results showed an increased risk of rebleeding but decreased mortality and thromboembolic complications in oral anticoagulant-resumed patients. As DOACs have completely different pharmacological features from VKA, extrapolating these results to DOAC-treated patients is not possible.

### 4.1. Recurrent Gastrointestinal Bleeding

In contrast with the previous meta-analyses [11,12], our results did not demonstrate significantly increased odds for recurrent GIB in the therapy resumption group.

As to the background of this controversial result, a better safety profile of DOACs could be an explanation. Data from three studies [17,39,40] observing the rebleeding rate after VKA or DOAC resumption are conflicted. In the study by Hernandez [40], patients resuming dabigatran had a significantly lower risk for major rebleeding (including mainly GI, but also intracranial and other sites) compared to patients resuming VKA (HR 0.42 95% CI 0.21–0.84 vs. 1.59 95% CI 1.10–2.22). At the same time, Candeloro [17] reported that restarting DOAC was associated with a slightly more elevated proportion of major bleeding than VKA resumption (17.4% vs. 15.6%). In a study by Rajan et al. [36], there was a similar hazard of rebleeding in the VKA and DOAC groups (HR 0.98 95% CI 0.75–1.28 and 0.82 95% CI 0.59–1.13, respectively).

Rebleeding is determined by many other factors that also have to be considered. Sources of the index bleeding were detailed in three included articles [37,38,39]. No remarkable differences in the location of bleeding sites were observed between the intervention and control groups. However, patients with more severe bleeding (higher rate of major bleeding, transfusion, and need for ICU admission) were significantly more likely to discontinue DOAC therapy. A study regarding oral anticoagulant-treated patients [17] found a significantly higher clinically relevant rebleeding rate in patients hospitalized with major index bleeding. Adequate endoscopic intervention and medical treatment positively impact rebleeding rate [43], although they are not described in the articles involved in our analysis.

Other well-known risk factors for gastrointestinal bleeding, such as additional antiplatelet therapy, were observed in the article by Sengupta et al. [37]. In the univariate Cox regression, aspirin or thienopyridine use after index bleeding was associated with a higher risk of rebleeding. However, their distribution was not reported between the intervention and control groups at discharge. Impaired renal function was not a risk factor for rebleeding in the article by Sengupta et al. [37], in contrast with the results from Candeloro et al. [17]. History of GIB was associated with a significantly higher risk of 90-day recurrent GIB.

### 4.2. Thromboembolic Complications

Previous VKA studies showed a clear benefit of anticoagulation resumption by decreasing thromboembolic complications [44,45,46,47]. In contrast, in the study by Sengupta et al. [37], no such reduction was proven. An explanation could be the shorter follow-up period and the possible underestimation of adverse events due to the data collection process (using medical claims data). In a conference abstract from Rajan et al. [36], the hazard of stroke and thromboembolism was lower in VKA and DOAC resuming patients compared to therapy withdrawal, although the difference was not significant (HR 0.79 95% CI 0.52–1.19 and 0.78 95% CI 0.48–1.28, respectively).

### 4.3. Mortality

In the study by Valanejad et al. [38], DOAC therapy resumption was not associated with mortality reduction, contrary to the conclusions of previous meta-analyses [11,12]. Serious imprecision could be observed because of very low event rates. Closely examining this study, we found a significantly higher number of patients with active cancer in the group of DOAC therapy resumption, which could impact the 12-month all-cause mortality.

### 4.4. General Considerations

To better understand the risk and benefits of DOAC resumption, the mortality of DOAC-related GI bleeding and thromboembolic complications associated with therapy discontinuation must also be revealed. The in-hospital or 30-day case-fatality rate of DOAC-related GIB varies between 1.6 to 7% [48,49], and DOAC-related major bleeding is 7.5% [50]. In contrast, the case fatality rate of stroke at 3 months is around 13%, while at 5 years, it rises to 46% [51]. In addition to mortality, the huge burden of long-term disability after stroke also has to be considered.

The optimal timing of anticoagulant therapy resumption remains unclear. Wide time variations could be observed between the included studies. In the study by Sengupta et al. [37], claims filled for DOACs in the first 30 days after hospital discharge did not show any association with gastrointestinal rebleeding or thromboembolic events. The resumption of DOACs in the first week from admission with index GI bleeding was not associated with significantly higher rebleeding or mortality rate in the study by Valanejad et al. [38]. In contrast, previous VKA studies showed a significant increase in rebleeding if therapy was restarted in the first week [44,45]. A recently published study by Candeloro et al. [17] observed an increased rebleeding risk in VKA- and DOAC-treated patients if the anticoagulation was resumed during the first week (11%). However, they found only a modest reduction in the risk of rebleeding if therapy was restarted after the second or third week (8% and 9%, respectively). The thrombosis rate was not dependent on the duration of the anticoagulation interruption. In a pilot study by Yamaguchi et al. [52], the authors investigated rebleeding, thromboembolic events, and mortality in patients with and without a short interruption (an average of 6.5 days) of antithrombotic agents (VKA, DOACs, and/or antiplatelets) after successful endoscopic hemostasis. No significant differences were found in the rebleeding outcome between the two groups. Only one ongoing randomized-controlled trial is investigating the safety and efficacy of DOAC resumption after GIB [53]. Therapy is restarted very early (within 24 h) or early (72–84 h) after successful endoscopic intervention in patients hospitalized with non-variceal upper GIB. There are no published preliminary results.

Due to insufficient data, we could not perform the planned subgroup analyses based on types of DOACs. In the study by Sengupta et al. [37], in both 3- and 6-month survival analyses, rivaroxaban proved to have the highest rebleeding rate compared to dabigatran and apixaban. No differences were found in thromboembolic complications among DOACs.

### 4.5. Strengths and Limitations

This is the first meta-analysis focusing only on DOACs to date. With the expanding consumption of DOACs, the topic is increasingly important. Our study was performed with the most up-to-date methodology, ensuring the highest quality, transparency, and reproducibility.

Considering the limitations of this work, we have to mention the low number of included studies and the low quality of evidence. There is a notable heterogeneity in methodology, populations, interventions, and outcomes analyzed. Due to the observational design of the included studies, a high risk of confounding can be observed. Patients with poor current status, more comorbidities, severe index bleeding, and more complementary therapy were more likely to discontinue DOAC therapy permanently.

The generalization of the results is problematic, while in most cases, the indication of therapy was atrial fibrillation, and the bleeding sources were not detailed enough.

### 4.6. Implications for Practice and Research

Our results offer a perspective that can mitigate the general fear of anticoagulant resumption after adverse events and decrease the unreasonably high proportion of DOAC therapy withdrawal. However, strong recommendations should not be provided based on the available data.

As gastrointestinal bleeding is a multifactorial disease, the impact of DOACs on rebleeding can be determined if all the other co-factors are taken into account or excluded with randomization. In future investigations, the following known confounding factors should be considered: bleeding source (location, type, number, size), endoscopic interventions and medical treatment used during the management of the index bleeding, additional medical therapy (antiplatelets, non-steroid anti-inflammatory drugs), DOAC effect modifying conditions (renal or hepatic impairment), and medications (CYP3A4/P-gp modifying drugs) as well as Helicobacter pylori infection in peptic ulcer disease. Furthermore, additional risk factors of rebleeding should be sought, and a novel score system should be established to estimate rebleeding risk. The optimal timing of DOAC therapy resumption could be safely investigated in patients without a high risk of rebleeding. There are insufficient data on the relationship between thromboembolic complications, mortality, and DOAC therapy resumption, although previous VKA studies proved a clear benefit of therapy restoration after GIB.

## 5. Conclusions

On the basis of our results, restarting DOAC therapy after a gastrointestinal hemorrhage may not be associated with a higher risk of GI rebleeding. However, the results must be interpreted with caution due to the low quality of evidence. Personal risk stratification must precede the decision-making of therapy resumption. There is an unmet need for further prospective data collection to assess the optimal timing of DOAC therapy resumption after GIB.

## Figures and Tables

**Figure 1 biomedicines-11-00554-f001:**
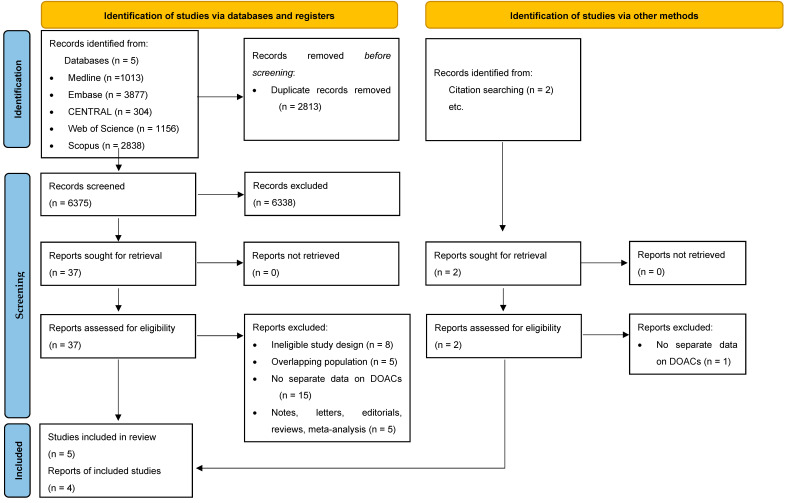
PRISMA 2020 flow diagram.

**Figure 2 biomedicines-11-00554-f002:**
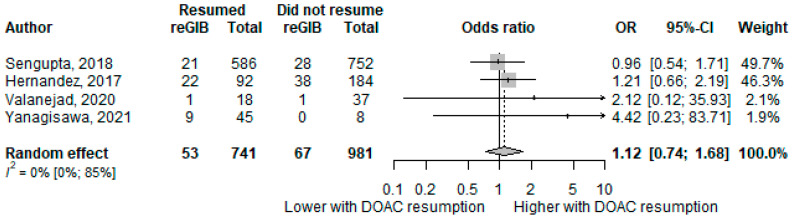
Forest plot representing the odds of rebleeding. Abbreviations: reGIB recurrent gastrointestinal bleeding, OR odds ratio, CI confidence interval [37,38,39,40].

**Figure 3 biomedicines-11-00554-f003:**
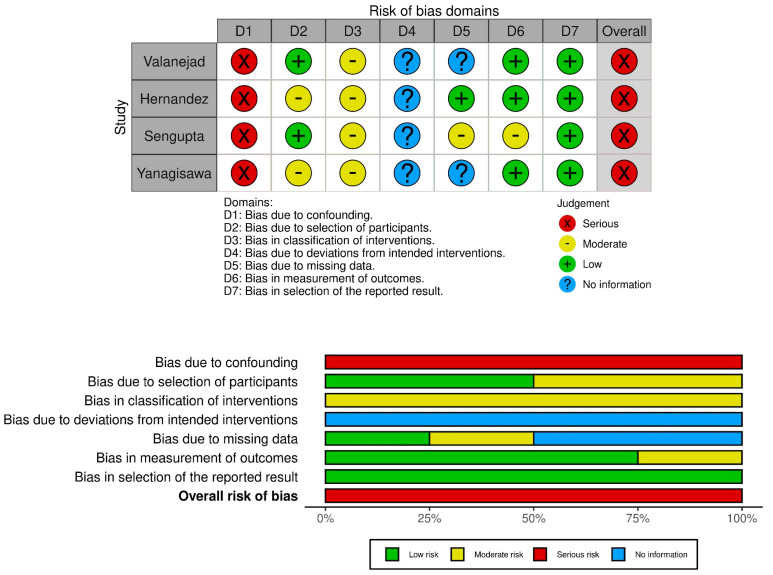
Risk of bias assessment (ROBINS–I tool) Outcome: recurrent gastrointestinal bleeding [37,38,39,40].

**Table 1 biomedicines-11-00554-t001:** Characteristics of included studies.

**Study**	**Design**	**Anticoagulant**	**Indication**	**Number of Patients**		**Age** **(Median, IQR)**		**Female (%)**	**Major GIB** **(%)**	**Intervention**	**Outcomes**	**Follow-Up Length (Months)**
				Resumed	Did Not Resume	Resumed	Did Not Resume					
Sengupta (2018) [37]	Retrospective cohort	Apixaban (n = 51, 4%) Rivaroxaban (n = 608, 45%) Dabigatran (n = 679, 51%)	AF	586	752	78 (70–83)	79 (71–84)	687 (51)	n/a	Resumption during follow-up (median 40 days; IQR: 17-88)	90-day/6 months hospital readmission with GIB or thromboembolic complications	6
Valanejad (2020) [38]	Retrospective cohort	Apixaban (n = 18, 31.6%) Rivaroxaban (n = 34, 59.6%) Dabigatran (n = 5, 8.8%)	AF, DVT, PE	37	18	75 (68–79)	74.5 (71.3–82.5)	31 (56)	37 (67) ^a^	Resume in ≤7 days from admission	90-day hospital readmission with GIB, 12 months mortality	12
Yanagisawa (2021) * [39]	Retrospective Cohort	Apixaban (n = 11, 20.8%) Rivaroxaban (n = 27, 50.9%) Dabigatran (n = 13, 24.5%) Edoxaban (n = 2, 3.8%)	AF	45	8	77	79.5	23 (43.3)	26 (46) ^b^	Resumption during follow-up (90% in 14 days)	Recurrent GIB or MACCE during follow-up	28 (10–44) ^d^
Hernandez (2017) * [40]	Retrospective cohort	Dabigatran (n = 276) (Warfarin)	AF	92	184	79.64 (8.67) ^c^	81.9 (7.63) ^c^	225 (67.3)	276 (100) ^a^	Resume in 3 months	Recurrent GIB during follow-up *	12

^a^—regarding ISTH criteria; ^b^—BARC type 3 or more; ^c^—(mean, SD), given for the whole DOAC-treated population regardless of bleeding site (80%GIB); ^d^—(median, IQR). * Note: the study by Hernandez et al. reported data from DOAC and VKA-treated patients with different site of major bleeding, while in the article of Yanagisawa et al. DOAC treated patients with different bleeding sites were included. In our study, only data from DOAC-related GIB patients were included. Abbreviations: AF atrial fibrillation, DVT deep vein thrombosis, GIB gastrointestinal bleeding, PE pulmonary embolism, n/a not available, MACCE major adverse cardiac and cerebrovascular events.

**Table 2 biomedicines-11-00554-t002:** Summary of Findings table.

Certainty Assessment							No. of Patients		Effect		Certainty	Importance
No. of Studies	Study Design	Risk of Bias	Inconsistency	Indirectness	Imprecision	Other Considerations	DOACs	No Therapy	Relative (95% CI)	Absolute (95% CI)		
Recurrent bleeding (assessed with: event)												
4	observational studies	very serious ^a^	not serious ^b^	not serious	serious ^c^	none	53/741 (7.2%)	67/981 (6.8%)	OR 1.09 (0.72 to 1.64)	6 more per 1000 (from 18 fewer to 39 more)	⨁◯◯◯ Very low	CRITICAL
All-cause mortality (follow-up: mean 12; assessed with: event)												
1	observational studies	serious ^d^	not serious	not serious	very serious ^c^	none	4/18 (22.2%)	4/37 (10.8%)	OR 2.36 (0.52 to 10.78)	114 more per 1000 (from 49 fewer to 458 more)	⨁◯◯◯ Very low	IMPORTANT
Thromboembolic event (follow-up: mean 3 months; assessed with: event)												
1	observational studies	very serious ^e^	not serious	not serious	not serious	none	16/586 (2.7%)	17/752 (2.3%)	OR 1.21 (0.61 to 2.42)	5 more per 1000 (from 9 fewer to 30 more)	⨁◯◯◯ Very low	CRITICAL
Majer adverse cardiac and cerebrovascular events (MACCE)												
1	observational studies	very serious ^e^	not serious	not serious	very serious ^c^	none	1/45 (2.2%)	3/8 (37.5%)	OR 0.037 (0.003 to 0.430)	353 fewer per 1000 (from 373 fewer to 170 fewer)	⨁◯◯◯ Very low	CRITICAL

CI: confidence interval; OR: odds ratio. Explanations: (a) small number of retrospective cohort studies, small sample size, serious risk of bias due to confounding (significant differences in baseline characteristics). (b) No statistically significant heterogeneity (I2 0%), overlapping confidence intervals. (c) Small sample size, low number of events, wide confidence intervals. (d) Retrospective cohort study, small sample size, serious risk of bias due to confounding (significant differences in baseline characteristics). (e) retrospective cohort study, serious risk of bias due to confounding (significant differences in baseline characteristics).

## Data Availability

Not applicable.

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
