# Peer review of "No Association between Gastrointestinal Rebleeding and DOAC Therapy Resumption: A Systematic Review and Meta-Analysis"

_biomedicines, 2023, doi:10.3390/biomedicines11020554_

Round 1

Reviewer 1 Report

I congratulate with the authors for the paper.

Actually there are recommendations for anticoagulation resumption after gastrointestinal bleeding (GIB), although data addressing this topic by direct oral anticoagulants (DOACs) treated patients is lacking. The authors try to to determine the safety and efficacy of restarting DOACs after GIB.

Studies that reported rebleeding, thromboembolic events, and mortality after restarting or withholding DOACs were selected. The systematic research was conducted in five databases  (MEDLINE, EMBASE, CENTRAL, Web of Science, Scopus). Random effect model was implemented to calculate pooled odds ratio (OR). ROBINS-I tool was used for risk of bias assessment, certainty of the evidence was evaluated with the GRADE approach. Results: 4 retrospective cohort studies (1,722 23 patients) were included in the meta-analysis. We did not find a significant increase in the risk of rebleeding in patients restarting DOACs after index GIB (OR=1.12; 95%CI: 0.74–1.68). The outcomes of thromboembolic events and mortality data were not suitable for meta-analytic calculations. Single studies did not show statistically significant differences in these. Data quality assessment showed a serious overall risk of bias and very low quality of evidence (GRADE D). The authors concluded that  DOAC resumption after a GIB episode may not elevate the risk of rebleeding. 

These conclusion are very important and interesting.

Author Response

 Dear Reviewer,
Many thanks for your review. We greatly appreciated your very valuable time invested in our manuscript.

Because of the better readability, we changed the orientation of figure 1 (PRISMA flow diagram) and table 1 (Characteristics of included studies) back. Orienting the figure vertically resulted loss of content. 

We also did one minor correction in the manuscript (see in reviewing panel).  

We hope that our revised manuscript will meet your expectations and you will find it suitable for publication. Please let us know if any further changes are required.

Again, thank you for your very valuable time invested in our manuscript.

Best regards,    Bálint Erőss

Reviewer 2 Report

This systematic review and meta-analysis is of high importance for clinicians. The topic is very attractive. Regarding the used methodology and the presented results, no suggestions for correction. The Discussion part is solid. My recommendation is to accept this paper.

Author Response

Dear Reviewer,
Many thanks for your review.

We did a minor correction in the manuscript (see reviewing panel). 

We also changed the orientation of figure 1 (PRISMA flow diagram) and table 1 (Characteristics of included studies) back to landscape, as portrait orientation resulted in a loss of content and difficulty in readability.   

We hope that our revised manuscript will meet your expectations and you will find it suitable for publication. Please let us know if any further changes are required.

Again, thank you for your very valuable time invested in our manuscript.

Best regards,

Bálint Erőss
